# Prenatal Sonographic Features of Ring Chromosome 15: A Case Report and Literature Review

**DOI:** 10.3390/diagnostics12040885

**Published:** 2022-04-01

**Authors:** Kuntharee Traisrisilp, Yuri Yanase, Krittaya Phirom, Theera Tongsong

**Affiliations:** 1Department of Obstetrics and Gynecology, Faculty of Medicine, Chiang Mai University, Chiang Mai 50200, Thailand; krittaya_p@docchula.com; 2Department of Obstetrics and Gynecology, Nakornping Hospital, Chiang Mai 50180, Thailand; yuletides@hotmail.com

**Keywords:** prenatal diagnosis, ring chromosome 15, ultrasound

## Abstract

Ring chromosome 15, a rare genetic disease, is very rarely prenatally diagnosed. We present a unique case of fetal ring chromosome 15 with ultrasound findings at 32 weeks of gestation including congenital diaphragmatic hernia, hypoplasia of the aorta with persistent left SVC, growth restriction, clubfeet and scoliosis. We also performed an analytical literature review of prenatal sonographic findings of the disease. This review suggests that ring chromosome 15 has a relatively specific sonographic pattern that could facilitate early detection. The specific sonographic features of ring chromosome 15 include fetal growth restriction, congenital diaphragmatic hernia, abnormal limb postures, cardiac defects, low-set ears and other less frequent, non-specific anomalies that can be identified in more than 50% of cases.

## 1. Introduction

Ring chromosome 15 is a very rare genetic disorder associated with a wide spectrum of phenotypes, namely prenatal and postnatal mild-to-severe growth failure, intellectual disability, facial dysmorphism, café-au-lait spots, cardiac defect, fertility problems and other minor dysmorphic stigmata [1,2,3,4,5,6]. Based on a literature search from 1966 to date, less than 100 cases have been documented and published. Most of them were postnatally diagnosed, either in childhood or adulthood, whereas prenatal diagnosis focusing on prenatal sonographic features of the disorder are very rarely published. To the best of our knowledge, only five cases of fetal ring chromosome 15 have been described based on prenatal sonographic findings [7,8,9,10,11]. The phenotypes of this syndrome are usually associated with serious morbidity, and it is lethal in some cases. Thus, prenatal diagnosis of this disorder is essential for clinical decision and management. Unfortunately, in spite of its importance, prenatal diagnosis is very rare. This might be associated with the existence of no obvious ultrasound findings in early gestation and no specific serum biomarkers warranting chromosome study as seen in fetal aneuploidy screening. To date the main challenge is how to facilitate early detection of this disorder. It is possible that most survivors had only subtle ultrasound findings at the time of screening or that such features were late-occurring. As mentioned above, most surviving cases of ring chromosome 15 have not been prenatally diagnosed despite the presence of a variety of structural anomalies. Thus, this study highlights the importance of sonographic features of fetal ring chromosome 15. Hopefully, pattern recognition with a high suspicion index based on some sonographic clues would facilitate early detection and offer more choices of management. Notably, although there have been scattered reports on prenatal diagnosis of the disease, a specific pattern has never been systematically reviewed or proposed. The objectives of this study are as follows: to present a unique case of ring chromosome 15, very rarely prenatally described, and to perform a literature review on fetal sonographic features of ring chromosome 15, which may be clinically used for pattern recognition, leading to early prenatal diagnosis.

## 2. Case Presentation

The case study was ethically approved by the institutional review board (Faculty of Medicine, Chiang Mai University) and written informed consent was provided. A 30-year-old Thai woman, gravida 3, para 2002, attended her first antenatal visit at a public hospital at 13 weeks of gestation. Her first two pregnancies were uneventful and both children were healthy. She had no known chronic medical diseases and no significant familial history of major disease or specific disorders. There was no consanguinity in the family and no teratogen exposure was noted. A maternal serum quadruple test for fetal Down syndrome screening showed a low risk for trisomy 21 (1:3377) (Alpha-fetoprotein: 0.66 MoM; beta-hCG: 0.24 MoM; Inhibin: 0.62 MoM; unconjugated estriol: 0.52 MoM). A fetal ultrasound at 13 weeks revealed normal structure and fetal biometry. Nuchal translucency was within the normal limit. An ultrasound anomaly screening at mid-pregnancy was not performed. At 32 weeks of gestation, an ultrasound examination was performed to evaluate fetal growth at a public hospital. A fetal anomaly was suspected and the patient was referred to a tertiary care hospital for further evaluation. Ultrasound showed a narrowing cisterna magna with intact cerebellar hemisphere and normal vermis; cardiac dextroposition (the apex pointing to the left side); left postero-lateral diaphragmatic hernia, containing the stomach and small bowel; and bilateral abnormal posture of both feet. (Figure 1). The stomach was located in the midline position of the thorax. A cytogenetic study by cordocentesis blood analysis was conducted. The karyotype showed mosaicism of 45,XX, del(15) [4]/46,XX, r(15) [26] from 30 metaphases (Figure 2). She was referred to our hospital at 35 weeks of gestation for proper prenatal and neonatal management. The ultrasound findings were the same as those mentioned above, with additional findings of coarctation of the aorta with retrograde flow, scoliosis at the thoraco-lumbar spine, bilateral club feet, and fetal growth restriction (estimated fetal weight of 1059 gm; 10th percentile: 1910 gm). The diaphragmatic hernia also contained the liver, and occupied most of the left side with a mediastinal shift, the lung-to-head ratio (LHR) was 0.75 (observed/expected LHR: 13.42–16.26%). After counseling, the couple chose to have expectant management. Spontaneous delivery occurred at 36^+4^ weeks of gestation and the fetus died during early labor. Finally the woman had a normal vaginal delivery, giving birth to a female stillborn baby (APGAR score 0 and 0 at 1 and 5 min, respectively) weighing 1490 gm. Postnatal findings (Figure 3) showed a normal umbilical cord; normal female genitalia; low-set ears; abnormal posture of the extremities, including fixed extension of both knees and single palmar crease of the left hand; external rotation of the right leg with severe talipes varus; external rotation of the left hip with external rotation of the left knee and leg with severe talipes vulgus. A postnatal X-ray confirmed the findings. Autopsy findings demonstrated a large diaphragmatic hernia containing the total left lobe of the liver, the stomach, the small bowel and large bowel as well as the appendix, spleen and pancreas, whereas the right lobe of the liver was located in the abdomen. The heart was displaced to occupy the greater part of the right thorax. The diaphragmatic defect was 3.3 cm × 2.2 cm at left postero-lateral portion and the residual lung tissue was tiny. The dextropositioned heart showed hypoplasia of the ascending aorta and aortic arch. Persistent left SVC was also identified and the anterior papillary muscle of the right ventricle had no chordae tendinae. The interventricular septum was intact. Scoliosis at the lower thoracic spine was noted and cytogenetic studies of the parents revealed normal results.

## 3. Literature Review

Publications on the prenatal diagnosis of fetal ring chromosome 15 were digitally searched for and comprehensively reviewed by the authors. All medical publications included in this review were obtained by digital search on the standard medical databases: PubMed, SCOPUS, and Web of Science, from 1966 to January 2022. Two authors independently assessed the title, abstract, and full text of the articles using the following key word: ring chromosome 15. The abstracts of the retrieved articles were manually screened to identify those focusing on prenatal sonographic features of the disorders. A total of 98 full-text articles were initially retrieved. After careful review, only six articles exactly involved fetal sonographic features of ring chromosome 15 and they were included in the review and analysis [7,8,9,10,11,12]. The data of baseline characteristics and ultrasound findings were extracted and pooled for analysis. Simple descriptive analysis was carried out and the data were presented as mean or percentage as appropriate, using the statistical package for the social sciences (SPSS) software version 26.0 (IBM Corp. Released 2019. IBM SPSS Statistics for Windows, Version 26.0, IBM Corp., Armonk, NY, USA). The results of the review analysis are presented in Table 1 and Table 2. Of the seven cases (including our case), the mean (±SD) maternal age is 30.9 + 3.0 years. The mean (±SD) gestational age at the time of sonographic diagnosis is 17.4 + 8.7 weeks. The prenatal sonographic features of ring chromosome 15 seem to be relatively specific. Fetal growth restriction is the most common finding, accounting for 83.3% of cases (excluding Manolakos’s case which was diagnosed at 11 weeks, too early for growth assessment). Congenital diaphragmatic hernia is also very common, found in 66.7%, followed by a thickened nuchal fold. In summary, based on this small series, the common prenatal sonographic features consistent with ring chromosome 15 are fetal growth restriction, congenital diaphragmatic hernia, thickened nuchal fold, abnormal limb postures and low-set ears. Cardiac defect is relatively common but the specific cardiac diseases vary (aortic hypoplasia and left SVC, ASD, etc.), making it difficult to specify the pattern. Importantly, facial dysmorphism is also common, but it is difficult for prenatal recognition due to 3D/4D ultrasound, which is more sensitive in detecting the dysmorphism, not being available in most centers.

## 4. Discussion

The new insight gained from this study is that fetal ring chromosome 15 seems to have a specific fetal phenotypic pattern, including specific sonographic features and other abnormalities not detected by ultrasound. Pattern recognition raising the possibility of ring chromosome 15 includes fetal growth restriction, congenital diaphragmatic hernia, abnormal limb postures, cardiac defects, low-set ears and other less frequent anomalies. Additionally, some abnormalities may be difficult to detect by prenatal ultrasound, especially facial dysmorphism or low-set ears. However, such abnormalities can probably be detected with high-resolution ultrasound in the near future. Note that low-set ears in ring chromosome 15, although usually not detected by routine anatomical screening, could be diagnosed by 3D/4D ultrasound as demonstrated by Britto et al. [11].

There have been scattered reports on several prenatal sonographic findings of ring chromosome 15, without a specific pattern. To the best of our knowledge, completely normal fetuses of rCh15 has never been described. However, based on a literature review including our case, we have analyzed and organized the findings to propose the sonographic features of ring chromosome 15 for pattern recognition and facilitation of early detection. Nevertheless, this analysis is based on a small sample size, and an accumulation of cases in the literature are still required to provide more confidence that the various findings are significantly associated with the disorder, rather than being co-incidental findings in a small series.

Unfortunately, the case presented here did not undergo cell-free fetal DNA analysis, nor molecular genetic sequencing to locate the exact region of deletion. Thus, findings from our case should be interpreted with caution. Additionally, it should be noted that our case had mosaicism such that there was a cell-line of 45,XX, del(15), although a minor component (4:30 metaphases), as in the case reported by Manolakos et al. [9]. However, the phenotypic pattern of our case is consistent with other previous reports usually involving r(15)(p11.1q26.3). The pattern of sonographic features proposed by our study is supported by previous molecular genetic studies as follows: Many studies have demonstrated that a critical region responsible for CDH is localized in 15q26.1–q26.2 [13,14,15,16], which is typically deleted in ring chromosome 15. Additionally, prenatal and postnatal growth deficiency is a common finding in cases of ring chromosome 15. This is consistent with the growing evidence supporting the loss of the insulin-like growth factor I receptor gene located in 15q26.3 [8,9,12,17], also absent in cases of ring chromosome 15. Thus, the 15q26.2–q26.3 microdeletion in fetuses with ring chromosome 15 is attributed to the deletion and haploinsufficiency of the IGF1R gene, probably determining the main clinical features of growth failure [18].

Additionally, this study demonstrates a unique finding that has been prenatally described, hypoplasia of the ascending aorta and the aortic arch and persistent left SVC. These findings are likely associated with ring chromosome 15 or are possibly incidental findings. Nevertheless, it is important that they should be accumulated in the literature for inclusion in future analysis. In future studies, we will underline the importance of molecular genetic testing such as aCGH to delineate specific regions of involvement. This may provide genotype-phenotype correlation information, since several abnormalities, such as aortic hypoplasia seen in our case, may be associated with unexpected complex rearrangements in ring chromosome 15. In a condition which has a variable clinical appearance, such as ring chromosome 15, the precise genotypic abnormality is necessary for predicting outcome and prognosis. Finally, note that, since this case had a cell line of chromosome 15 deletion in a minority of the cell population, this may hypothetically be associated with cardiac defect in this case, although such a relation has never been described.

## 5. Conclusions

We present a unique case of fetal ring chromosome 15. We have also attempted to delineate prenatal sonographic features of the disorder by an analytical review of the scattered case reports available. The typical sonographic features include fetal growth restriction, congenital diaphragmatic hernia, abnormal limb postures, cardiac defects, low-set ears and other less frequent, non-specific anomalies that can be identified in more than 50% of cases.

## Figures and Tables

**Figure 1 diagnostics-12-00885-f001:**
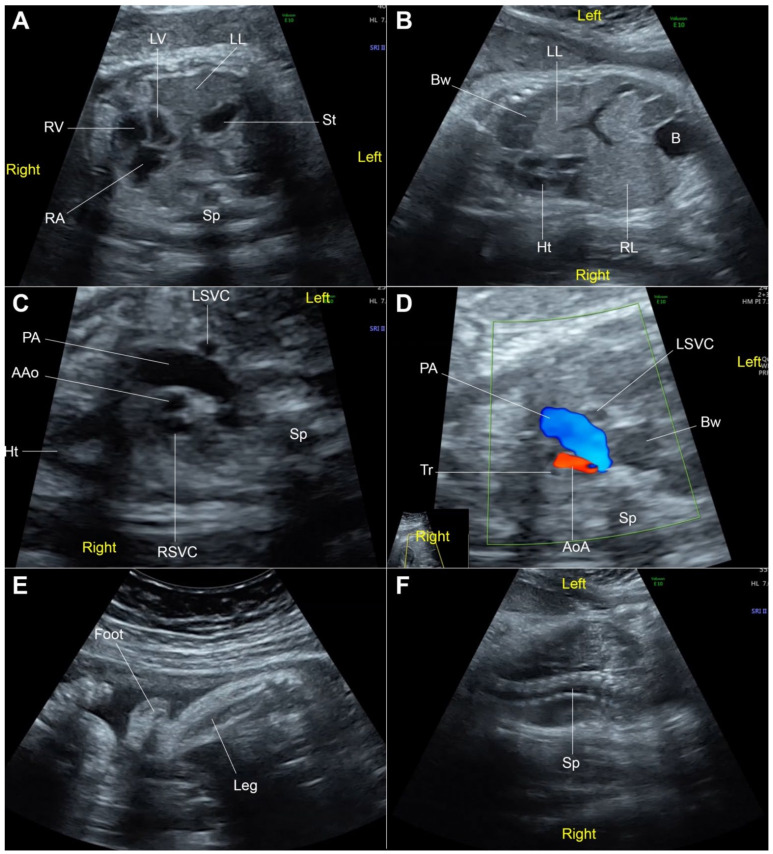
Fetal ultrasound at 32 weeks of gestation; (**A**) Cross-section at four-chamber view: the heart in the right chest with left-right disproportion, stomach (St) and liver (left lobe: LL) located in the left chest; (**B**) Coronal view of the trunk: the heart (Ht) in dextroposition, left lobe of the liver and part of portal vein located in the left chest; (**C**) Three-vessel view: small ascending aorta (AAo) and left superior vena cava (LSVC); (**D**) Three-vessel and trachea view: retrograde flow in the small aortic arch (AoA); (**E**) Talipes-equinovarus; (**F**) Scoliosis (B: bladder; Bw: bowel; LV: left ventricle; PA: pulmonary artery; RA: right atrium; RL: right lobe of the liver; RSVC: right superior vena cava; RV: right ventricle; Sp: spine; Tr: trachea).

**Figure 2 diagnostics-12-00885-f002:**
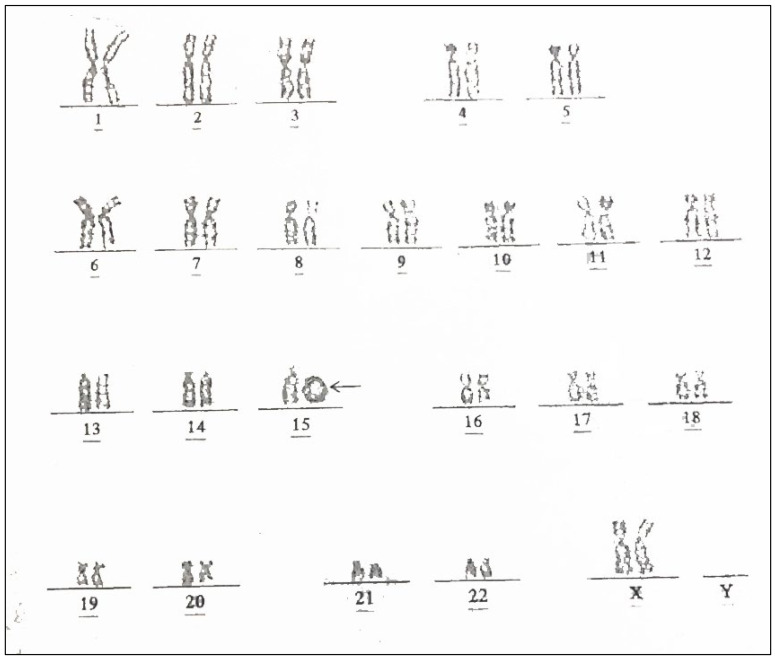
Karyotype of the fetus from the present study shows a ring chromosome 15; 46,XX, r(15) (arrow).

**Figure 3 diagnostics-12-00885-f003:**
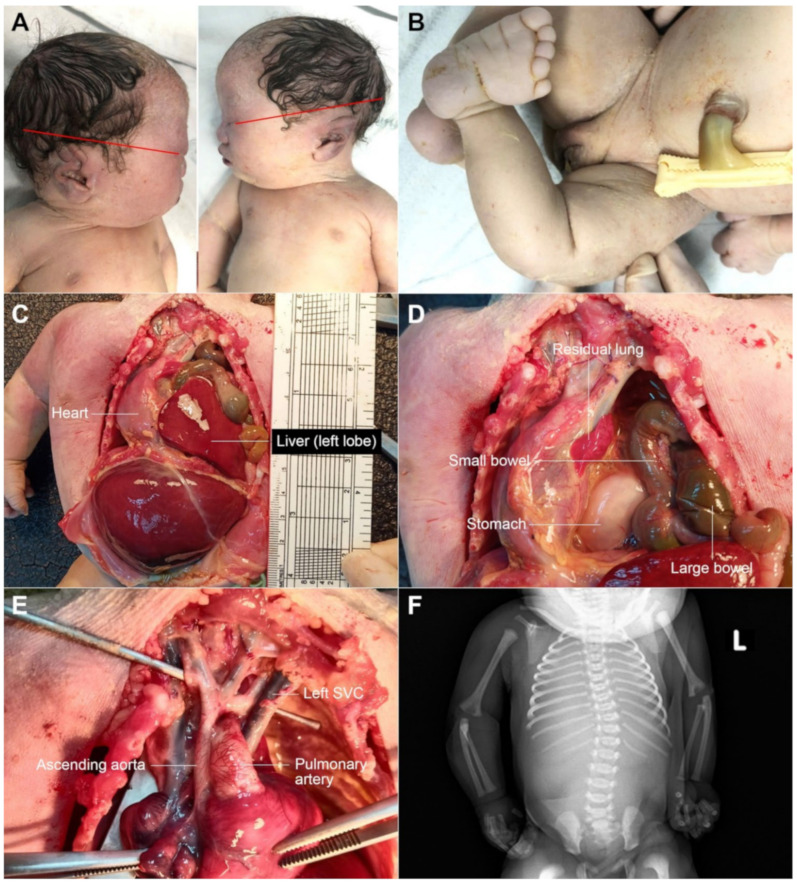
Postnatal findings; (**A**) low-set ears; (**B**) clubfoot; (**C**,**D**) diaphragmatic hernia; (**E**) small ascending aorta and aortic arch and left SVC; (**F**) scoliosis (L: left side).

**Table 1 diagnostics-12-00885-t001:** Sonographic features of 7 fetuses with ring chromosome 15.

References	Maternal Age (yr)	GA at Diagnosis	Structural Abnormalities	Outcomes
Liu et al. [7] (2001)	27	20	Increased nuchal fold thickness FGR	TOP
Glass et al. [12] (2006)	34	16	Increased nuchal fold thickness FGR Oligohydramnios Single umbilical artery Facial dysmorphism ASD Limb anomalies	Delivery
Hatem et al. [8] (2007)	27	18	FGR Oligohydramnios Facial dysmorphism CDH Polycystic kidneys	TOP
Manolakos et al. [9] (2009)	36	11	(Not seen at 11 weeks) Post-abortal autopsy: CDH Polycystic kidneys Facial dysmorphism Limb anomalies Dolichocephaly	TOP
Tan et al. [10] (2012)	33	19	Dandy-Walker malformation Hydrops fetalis Low-set ears High maternal hCG levels	TOP
Britto et al. [11] (2014)	31	19	Increased NT (13 week) FGR CDH Low-set ears	Delivery (36 weeks) Stillbirth
Present case (2022)	30	32	FGR CDH Low-set ears Hypoplastic aorta	Delivery (36 weeks) Stillbirth

**Table 2 diagnostics-12-00885-t002:** Common prenatal sonographic features of 7 fetuses with ring chromosome 15 (not included are the abnormalities unlikely to be detected by ultrasound screening).

Sonographic Features	Number (n/N)	Percent
Fetal growth restriction	5/6	83.3
Congenital diaphragmatic hernia	4/6	66.7
Thickened nuchal fold	4/7	57.1
Abnormal extremities (postures)	3/6	50.0
Low-set-ears	3/6	50.0
Oligohydramnios	2/5	40.0
Congenital heart disease	2/5	40.0
Others (single umbilical artery, abnormal spine etc.)	4/7	57.1

## Data Availability

The data of this report are available from the corresponding authors upon request.

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
