# Peer review of "Prenatal Sonographic Features of Ring Chromosome 15: A Case Report and Literature Review"

_diagnostics, 2022, doi:10.3390/diagnostics12040885_

Round 1

Reviewer 1 Report

You presented us an extremely interesting and very rare case. I would like to know if the fetus stopped evolving before the onset of the labor or during the labor. 

Author Response

Reviewer: 1 (highlighted in green)

Comments and Suggestions for Authors

You presented us an extremely interesting and very rare case. I would like to know if the fetus stopped evolving before the onset of the labor or during the labor.

Response: The fetus died during early labor. In revised MS, we provide this information in the “Case Presentation” section, as highlighted.

Reviewer 2 Report

This is a intriguing case report and literature review about a rare genetic condition leading to multiple fetal anomalies. I wish to congratulate the authors for their work and research.

Just a few observations:

  • Is this a systematic or narrative literature review? I would specify the nature of the reviewing process and potentially add a strobe flowchart to depict how the final case reports were selected.
  • Are there any reports of cases with ring chromosome 15 findings that did not have a correlation with fetal anomalies? If not, I would specify this in the text.
  • I appreciate the paragraph in the discussion that mentions the importance of the mosaicism finding in the genetic report. Is chrom 15 deletion related in any way to cardiac abnormalities, which the authors highlight as their unique finding in this case?

Once again, a very interesting case report and literature review.

Author Response

Reviewer: 2 (highlighted in red)

Comments and Suggestions for Authors

This is a intriguing case report and literature review about a rare genetic condition leading to multiple fetal anomalies. I wish to congratulate the authors for their work and research.

Just a few observations:

Is this a systematic or narrative literature review? I would specify the nature of the reviewing process and potentially add a strobe flowchart to depict how the final case reports were selected.

Response: This is a simple systematic review which need no strobe flow chart, we systematically searched and collected consecutive cases available on standard database and pooled all cases and performed simple analysis to determine frequencies of certain abnormalities. The search detail and analysis /statistical technique are described in “Literature Review” section.

Are there any reports of cases with ring chromosome 15 findings that did not have a correlation with fetal anomalies? If not, I would specify this in the text.

Response: We mention this statement in “Discussion” now, in the second paragraph of “Discusion”.

I appreciate the paragraph in the discussion that mentions the importance of the mosaicism finding in the genetic report. Is chrom 15 deletion related in any way to cardiac abnormalities, which the authors highlight as their unique finding in this case?

Response: In revised MS, we add this comment at the end of  “Discussion”, before “Conclusion” section

Once again, a very interesting case report and literature review.

Reviewer 3 Report

Dear Authors,

I have read your manuscript with pleasure and intelectual satisfaction. It is well written, including literature review and original prospective suggestions for prenatal ultrasound diagnosis of ring chromosome 15.

However I have some questions and suggestions to improve the paper.

1/ In the presentation readers should have a chance to see the report of 1 st trimester screening with US findings and measurements esp. NT, tricuspid flow, DV wave pattern and PI (any fotos from the examination?) and biochemical markers (PAPP-A, betaHCG, PLGF, others) expressed in direct values and in MoMs without just labelling as normal. Was the sonographer FMF and/or ISUOG certified?

2/ Lack of microarrays (aCGH) in this case is a limitation you mentioned in Discussion. But since cell free fetal DNA analysis becomes more available for pregnant women we can expect more frequent detection of ring chromosome 15 provided there are specific genetic disturbances. The subject could be mentioned in your Discussion.

3/ Following sentence 178-180 needs to be corrected:

This can be provide genotype-phenotype correlation information. Since several abnormalities like aortic hypoplasia in our case might be associated with unexpected complex rearrangements in ring chromosome 15. 

4/ You may want to present also all other phenotypic features which are currently unlikely detectable by ultrasound but could be such in the future.

5/ The case

Author Response

Reviewer: 3 (highlighted in blue)

Comments and Suggestions for Authors

I have read your manuscript with pleasure and intelectual satisfaction. It is well written, including literature review and original prospective suggestions for prenatal ultrasound diagnosis of ring chromosome 15.

However I have some questions and suggestions to improve the paper.

1/ In the presentation readers should have a chance to see the report of 1 st trimester screening with US findings and measurements esp. NT, tricuspid flow, DV wave pattern and PI (any fotos from the examination?) and biochemical markers (PAPP-A, betaHCG, PLGF, others) expressed in direct values and in MoMs without just labelling as normal. Was the sonographer FMF and/or ISUOG certified?

Response: NT was measured by MFM specialists (certified by Thai Board MFM training), but not ISUOG certified. We are sorry that we could not provided the information on first trimester US exams since this was performed in other community hospital. Also first trimester biochemical markers were not investigated. However, we provided biochemical markers of Quad test in the second trimester, as highlighted in blue in “Case Presentation”

2/ Lack of microarrays (aCGH) in this case is a limitation you mentioned in Discussion. But since cell free fetal DNA analysis becomes more available for pregnant women we can expect more frequent detection of ring chromosome 15 provided there are specific genetic disturbances. The subject could be mentioned in your Discussion.

Response: This is mentioned as a limition, as suggested, as highlighted in the third paragraph of “Discussion”.

3/ Following sentence 178-180 needs to be corrected:

This can be provide genotype-phenotype correlation information. Since several abnormalities like aortic hypoplasia in our case might be associated with unexpected complex rearrangements in ring chromosome 15.

Response: The sentences are corrected as suggested, as highlighted.

4/ You may want to present also all other phenotypic features which are currently unlikely detectable by ultrasound but could be such in the future.

Response: The statement is now added as suggested, as highlighted, in the first paragraph of “Discussion”.

5/ The case -